# A Review of Antecedents and Effects of Loyalty on Food Retailers toward Sustainability

Yating Tian [1,*] and Qeis Kamran [2]

[1] Department of Design Production & Management, University of Twente, Drienerlolaan 5, 7522 NB Enschede, The Netherlands
[2] International School of Management GmbH, Faculty of International Management, Otto-Hahn-Straße 19, 44227 Dortmund, Germany; qeis.kamran@ism.de
* Correspondence: y.tian-1@utwente.nl

**Abstract:** With the increase in consumer awareness of sustainability and diversified retailer brands, the conceptualizations and dimensions of brand loyalty are changing. Existing research studies have focused on traditional constructs and measurements to explain new phenomena in the food retail sector but ignored the environmental and social effects on consumers' attitudinal and behavioral loyalty. This study entails an extensive and structured review of definitions, taxonomy, dimensions, and measurements of loyalty within a food marketing context. With an additional emphasis on the notion of sustainability, it provides a perspective theory synthesis that integrates all testified antecedents of all types of loyalty to emphasize a trend of sustainability beyond brand scope, whereby sustainability values create loyalty. A systematic literature review and qualitative analysis methods were used to identify the relevant literature. The studies that qualified for inclusion were those that reported (1) research methods, (2) dimensions of brand loyalty, (3) knowledge of sustainability factors, and (4) organic marketing. This paper summarizes and compares the key constructs and measurements of loyalty to retailers. The results show inconsistencies in relation to two important attitudinal dimensions, namely, brand satisfaction and brand value. Although loyalty towards product brands, loyalty toward service organizations, store loyalty, and retailer loyalty have been studied in recent decades by marketing academics, little attention has been paid to clarifying their role in food retailing, especially regarding whether the established dimensions are relevant in conceptualizing consumer loyalty in sustainability based on organic food marketing. The theoretical implications are discussed in association with the research gap between loyalty dimensions and sustainability values, as well as multidimensional measurements development. The practical implications of this review are important for food retailers and organic food marketers that can meet the satisfaction and retain consumers' loyalty by providing organic and sustainable products and improving related service quality involving environmental consequences and social well-being.

**Keywords:** brand loyalty; food retailing; sustainability; organic marketing; values

## 1. Introduction

Loyalty is one of the most important assets of a corporate brand. Research in this area has examined evolutionary marketing activities and the relationships with customers. The growth of customer-centric marketing has occurred in the product-, market-, and customer-oriented phases [1]. Brand loyalty has prospered due to the strong marketing inputs, but existing research studies have only explored this concept using sole dimensional measures, for example, in terms of a behavioral dimension for an earlier time period. Therefore, consumer loyalty deserves recognition as a multi-dimensional construct [2]. In the food market, consumers are increasingly concerned about the environment and the realization of a sustainable society. Their interest is aroused by organic, pro-environmental, and sustainable products. Therefore, the sustainability-oriented marketing notion has

increasingly proliferated in food retailing and academic research [3,4]. A challenge for food retail corporations is implementing effective brand-focused marketing strategies over the long term to consolidate consumer loyalty. Furthermore, corporate executives evaluate the essential performance of their marketing strategies to better understand what antecedents are important for ensuring consumer loyalty in food marketing and how to measure it.

The current specifications of predictors to consumer loyalty to food retailers have a disputable theoretical background. Based on the extant literature about antecedents and constructs of loyalty, combined with the contemporary research background of integrating sustainability in marketing and the practical sustainable business, the current theoretical background, and new challenges are visualized in Figure 1.

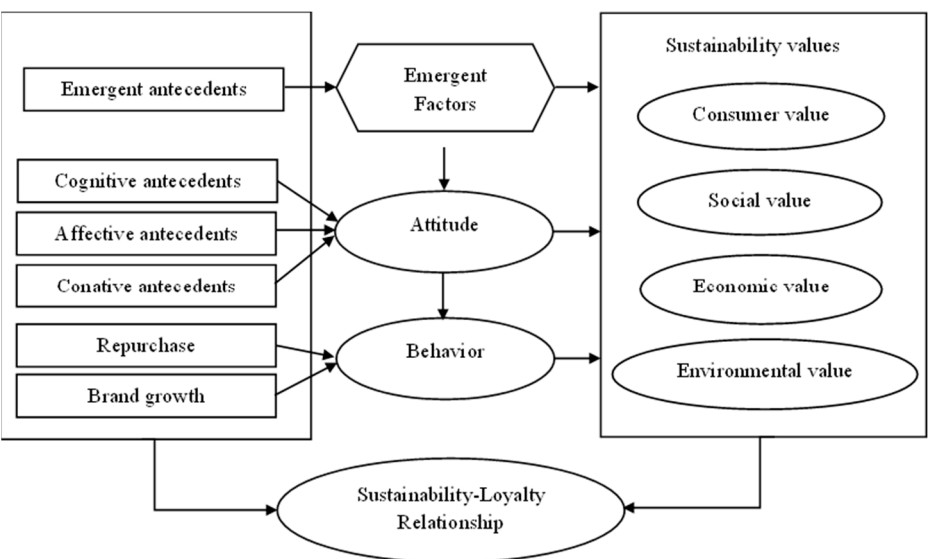

**Figure 1.** Theoretical Background and New Challenges.

Therefore, this study answers the following questions: (1) What are the antecedents of loyalty in relation to marketing? (2) What dimensions have newly evolved pertaining to food retailers' sustainable marketing? (3) How should this loyalty be discerned? To this end, a systematic literature review was conducted to identify valid dimensions and measures of consumer loyalty for food retailers selling sustainable food products.

The term "brand loyalty" is primarily used when describing product-oriented brands. There are a variety of definitions in the consumer loyalty literature, such as the earlier mainstream concept using the following six criteria: (1) the biased (2) behavioral response (3) expressed over time (4) by some decision-making unit (5) with respect to one or more alternative brands (6) which is a function of psychological processes [5–7]. This approach to defining brand loyalty has been deemed inadequate for explaining the heterogeneity of consumers' characteristics. Moreover, measurements of the behavioral response "purchase or repurchase" may be individual for reasons other than brand switching behavior [8–11].

Subsequently, the definition expanded to service loyalty, which has been typically derived from service organizations and developed in the market orientation phase. This concept has extended loyalty to the brands of organizations that provide intangible products [12]. The dimensions involve the following composite constructs: (1) behavioral loyalty, for example, typical purchase response [13]; (2) attitudinal loyalty, for example, consciousness and intentions [14]; and (3) cognitive loyalty, for example, brand preference underlying psychological commitment [15]. This theory is consistent with Dick and Basu's [16] combination model of relative attitude and repeat patronage.

The three central constructs of brand relationships pertaining to customer brand loyalty are attachment, trust, and identification, as demonstrated by Diallo et al. [17]. These constructs influence the relationships between brand image dimensions and the dimensions

of loyalty that incorporate cognitive, affective, and normative aspects. These dimensions can be categorized into attitudinal and socio-psychological attributes.

However, consumers' purchase intentions and their perception of retailer brands may be related to some emergent factors, for example, whether the brand is dedicated to values of sustainability and food safety, and public food policy as well. The core values of sustainability are economic, environmental, and social values [18–20]. Retailers have incorporated this practice into organic, healthy, and functional food marketing. This approach differs from other established tactics and induces new dimensional loyalty toward retailer corporations attitudinally, which may apply not only to tangible products but also to intangible brand assets.

Therefore, this study is motivated by the fact that there is a gap between defining consumer loyalty toward sustainability in marketing research and providing advanced measurements in the food business practice, which is still an open problem. Consumers are increasingly concerned with sustainability issues involving environmental consequences, social influences, and consumer well-being. Retailers are challenged to retain consumers and strengthen loyalty by reasonable input decisions for substantial profits.

This study is significant to the various stakeholders: (1) Food retailers—this review will help them determine the marketing inputs toward sustainability values based on organic products and related services to retain consumer loyalty. (2) Public policymakers— this study will hopefully enlighten associated governmental bodies regarding the necessity to enhance social well-being by authorizing appropriate organic control organizations, as well as motivating retailers to be members of organic associations. (3) Organic food manufacturers/producers—this review explains that organic products can create sustainability values for both retailers and their own brands by cooperation, which would be beneficial for organic food manufacturers/producers. The objective is to understand essential loyalty definitions and constructs of the concept of sustainability for enhancing consumer loyalty to food retailers. In comparison to previous studies, the novelty of this paper lies in the proposal of redefining loyalty based on sustainability values and integrating the multidimensional measurements for sustainable business in the food retail sector.

Increasing attention is paid to retailer loyalty in the marketing literature in various contexts [21,22], given its importance in retailing. However, we observed inconsistent conceptualizations and findings in these contexts. Moreover, the question of whether valid loyalty measurements are essential for retailers in terms of sustainability based on organic and sustainable marketing remains unexplored, calling for an emerging research agenda. Hence, the analysis starts with the chronological academic definitions of loyalty. Next, we evaluate the major types of methods used in related studies. Finally, conclusions are drawn and implications for future research directions for the managerial application of food retailers' loyalty measurements are provided.

## 2. Materials and Methods

This study uses a hybrid narrative review approach, which falls into a systematic review category, as it adopts two categories—theory and constructs—in the models. The examination of theory, context, and method (TCM) is conducted from an integrated view. Therefore, the TCM framework was developed based on previous review studies [23,24]. As recommended by the PRISMA statement, we chronologically investigated the existing body of loyalty research between 1961 and 2021 in major journals in the Web of Science database in the fields of business, retail brand management, and marketing. To ensure a high-quality analysis leading to a more integrated review, we started with broad keywords, such as loyalty, brand, and marketing, which implicate any type of discussion about these concepts and cover academic studies without omitting relevant and valuable information. Then, we focused on mainstream loyalty conceptualizations and their taxonomy; the selection criteria of keywords such as brand loyalty, service loyalty, retailer loyalty, consumer loyalty, organic marketing, and sustainability were specified. The selected studies were those that (1) elaborated on the definitions of loyalty; (2) conceptualized the dimensions of

loyalty; (3) provided measurement models and hypothesis test analysis; (4) reported the relationship directions of the effects; and (5) yielded statistically significant constructs, indicators, and predictors. A stepwise systematic desk search was performed and 117 studies were finally selected. Figure 2 shows a flow diagram of the review.

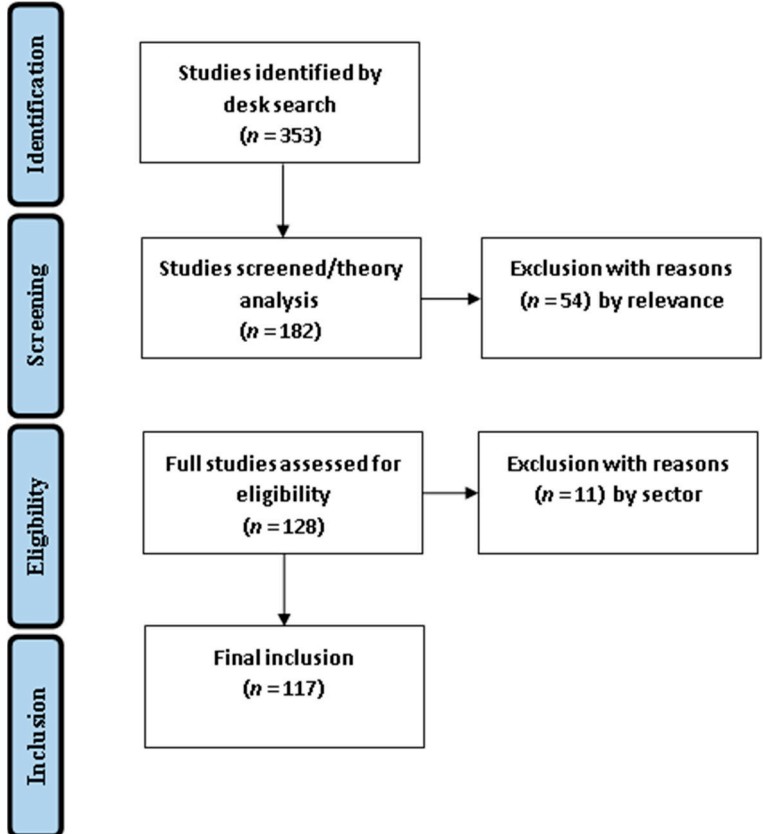

**Figure 2.** Method.

The first step was to search for all the articles concerning the original discussion on brand loyalty. The second step was to identify if the study cases included the dimensions concerning the conceptualization of brand loyalty and to analyze the theories, namely, examine which elements had been applied in each theory. The third step involved screening and extending the evolved loyalty concepts within market-oriented industries. The fourth step distinguished between general and sectorial loyalty, such as retailer loyalty. The fifth step identified unexplored loyalty themes in the retailing field. The final step focused on selecting the studies to be analyzed. We found a large amount of literature on loyalty. However, some of the identified studies [25,26] were screened out, owing to limitations relating to relevance, topic directions, and results. According to the eligibility test, we excluded studies from the fashion, energy, politics, luxury, and banking industries [27,28]. As this research study focuses on food retailer loyalty, especially at the sustainability level, which has been hitherto less explored, the final sample size of 117 was the result of rigorous selection. Appendix A shows the review steps with the keywords used.

## 3. Literature Review

Loyalty is a major source of brand equity, as it refers to retaining customers; the authors of [29–31] demonstrated an interactive relationship in this regard. They argued that brand equity is an important antecedent of customer loyalty, while other researchers hold that the direction of the relationship is the other way around, in that customer loyalty is an antecedent of brand equity [30]. Here, we provide a comprehensive discussion of definitions and dimensions, antecedents and determined constructs, design of measures,

and sustainability issues in relation to loyalty that have not been analyzed in depth in the literature.

### 3.1. Definitions

Based on the antecedents and components of loyalty, as well as the derived relational effects mentioned in the literature, we found that antecedents vary with the taxonomy of loyalty. In terms of distinctive characteristics, attributes of objectives, target groups of business activities, and type of industry, loyalty can be categorized into brand, service, store, and retailer loyalty [32]. Table 1 summarizes these definitions and dimensions from various perspectives.

**Table 1.** Definitions and dimensions of loyalty.

| Study | Attribute/Type | Dimension | Focus | Definition |
|---|---|---|---|---|
| Cunningham [11] | Customer loyalty to store (including chains) and brand | Behavioral | Repurchase | " . . . important is the proportion of a family's total food purchases that are made in any one particular store. This proportion . . . describes family's loyalty to any given store or combination of stores." (p. 128) |
| Tucker [13] | Brand loyalty | Behavioral | Choice | " . . . is conceived to be simply biased choice behavior with respect to branded merchandise . . . " (p. 32) |
| Jacoby and Kyner [6] | Consumer loyalty | Behavioral | Repurchase | " . . . is first distinguished from simple repeat purchasing behavior and then conceptually defined in terms of six necessary and collectively sufficient conditions . . . " (p. 1) |
| Sheth and Park [2] | Brand loyalty | Emotive, evaluative, and behavioral | Tendency | " . . . a positively biased tendency contains three distinct dimensions . . . the first dimension is the emotive tendency toward the brand . . . the second dimension is the evaluative tendency toward the brand . . . the third dimension is the behavioral tendency toward the brand . . . " (p. 450) |
| Jacoby and Chestnut [5] | Brand loyalty | Behavioral | Purchase | " . . . biased behavioral response, expressed over time, by some decisions-making unit, with respect to one or more alternative brands out of a set of such brands and is a function of psychological processes . . . " (p. 80) |
| Gremler and Brown [12] | Service loyalty | Behavioral, attitudinal, and cognitive | Satisfaction, switching costs, interpersonal bonds | " . . . is the degree to which a customer exhibits repeat purchasing behavior from a service provider, possesses a positive attitudinal disposition toward the provider and considers using only this provider when a need for this service arises . . . " (p. 173) |
| Bloemer and de Ruyter [21] | Store loyalty | Attitudinal and behavioral | Store satisfaction and store image | " . . . the repeat visiting behavior based on a maximum amount of commitment . . . " (p. 500) |
| Wallace et al. [22] | Retailer loyalty | Attitudinal and behavioral | Satisfaction | " . . . as the customer's attitudinal and behavioral preference for the retailer when compared with available competitive alternatives . . . " (p. 251) |
| Schultz and Block [33] | Brand sustainability | Organizational performance | Average growth rate (AGR) and net promoter score (NPS) | " . . . brands have some type of 'sustainable' quality, that is, they grow and evolve over time, there is increasing evidence that brands, such as other corporate resources, can decline and fail if not properly managed . . . " (p. 343) |

Table 1 shows the definitions and dimensions of the distinct types of loyalty from several pathbreaking studies. This table is based on a chronological evolutionary map of loyalty, which moves from product-concentrated to market-focused, to service-dependent, and to sustainability-provoked research studies. These studies have paved the way for understanding what antecedents, dimensions, and components can construct loyalty. Although they provide different perspectives, the converging points are customers or consumers.

*3.2. TCM Framework*

According to Chaudhary et al. [23], the TCM framework provides a clear understanding of how a theory has evolved historically from different perspectives. Moreover, the influential factors provide a mainstream research trend. Considering that journal articles published earlier have a higher citation probability, the papers ranked by the number of citations divided by the number of years since publications were used as "citations per year" [34]. Table 2 overviews the selected studies that share higher influential factors (cited over 10 times per year) according to SSCI, as well as mainstream theories and models, data collection methods, and the context.

**Table 2.** TCM framework.

| Study | Total Citations | Citations per Year | Theories and Models | Data Collection Methods | Context |
|---|---|---|---|---|---|
| Morgan and Hunt [35] | 8602 | 330.85 | Commitment Theory/ Rival Model | Survey | United States |
| Dick and Basu [16] | 3978 | 153.00 | Customer Loyalty/Attitude/Behavior | Conceptual | International |
| Chaudhuri and Holbrook [36] | 3099 | 163.10 | A Model of Brand Loyalty and Brand Performance | Survey | United States |
| Brakus et al. [37] | 1650 | 149.96 | Brand Experience Dimensions/Four-Factor Model | Experiential Brands | International |
| Boulding et al. [38] | 1432 | 53.04 | Behavioral Process Model | Experiment/Survey | United States |
| Caruana [39] | 1081 | 60.06 | Service Loyalty/ Mediational Model | Questionnaire Mailings | Malta |
| Reichheld and Schefter [40] | 1024 | 51.20 | E-Loyalty | Conceptual | United States |
| Thomson et al. [41] | 991 | 66.07 | Emotional Attachments to Brands | Survey | International |
| Anderson and Srinivasan [42] | 823 | 48.41 | Moderated Effect | Survey | International |
| Kim and Ko [43] | 766 | 95.73 | Structural Equation Model | Survey | Korea |
| Bloemer and de Ruyter [21] | 723 | 32.85 | Latent Satisfaction and Loyalty | Survey | Switzerland |
| Bloemer and Kasper [44] | 708 | 28.34 | Satisfaction—Loyalty Theory | Questionnaire | The Netherlands |
| Delgado-Ballester and Munuera-Alemán [45] | 595 | 31.30 | Overall Satisfaction and Loyalty | Interview | Spain |
| Gremler and Brown [12] | 575 | 23.98 | A Model of Service Loyalty | Interview | United States |
| Homburg and Giering (2001) | 537 | 28.26 | Satisfaction—Loyalty Theory | Survey | Germany |
| Sirohi et al. [46] | 529 | 24.05 | Consumer Perceptions and Store Loyalty | Phone Interview | United States |
| Jacoby and Kyner [6] | 514 | 10.94 | Brand Loyalty and Repeated Purchase | Experiment | United States |
| Uncles et al. [47] | 509 | 29.91 | Customer Loyalty | Conceptual | International |
| Bloemer et al. [48] | 476 | 22.65 | Service Loyalty | Interview | Belgium |
| Fullerton [49] | 455 | 26.79 | Commitment—Loyalty Theory | Experiment | Canada |
| Reichheld [50] | 452 | 16.74 | Employee Loyalty | Conceptual | United States |
| Chintagunta et al. [51] | 448 | 15.45 | Logit Model | Panel Data | United States |
| Corstjens and Lal [52] | 379 | 18.95 | Game Theory | Panel Data | International |
| Vlachos et al. [53] | 366 | 33.27 | Consumer Trust | Phone Interview | Greece |
| Jones and Suh [54] | 329 | 16.45 | Full/Partial Mediation Model and Moderation Model | Survey | United States |
| Tellis [55] | 320 | 9.99 | Rival Models | Scanner Records | United States |
| Agustin and Singh [56] | 318 | 21.20 | Structural Equation Model | Survey | United States |
| Pullman and Gross [57] | 303 | 18.94 | Latent Path Model | Survey | United States |
| Pivato et al. [58] | 290 | 24.17 | Trust and Brand Loyalty | Survey | EU |
| Wallace et al. [22] | 275 | 17.20 | Customer Retailer Loyalty | Survey | United States |

**Table 2.** *Cont*.

| Study | Total Citations | Citations per Year | Theories and Models | Data Collection Methods | Context |
|---|---|---|---|---|---|
| Evanschitzky and Wunderlich [59] | 272 | 19.43 | Four-Stage Loyalty Model | Survey | Germany |
| Gommans et al. [60] | 266 | 14.00 | The E-Loyalty Framework | Conceptual | International |
| Reichheld and Schefter [40] | 249 | 12.46 | E-Loyalty | Conceptual | United States |
| Mascarenhas et al. [61] | 245 | 17.53 | Total Customer Experience Approach | Conceptual | International |
| Palmatier et al. [62] | 240 | 18.44 | Salesperson-Owned Loyalty | Survey | United States |
| Taylor et al. [31] | 232 | 14.50 | Behavioral and Attitudinal Loyalty | Survey | United States |
| Iglesias et al. [63] | 226 | 22.61 | Brand Experience and Brand Loyalty | Survey | Spain |
| Ailawadi et al. [64] | 221 | 18.42 | Behavioral Loyalty | Panel Data | The Netherlands |
| Evanschitzky et al. [65] | 210 | 15.00 | Attitudinal and Behavioral Loyalty | Survey | Western Europe |
| Bandyopadhyay and Martell [66] | 203 | 15.64 | Attitudinal and Behavioral Loyalty | Survey | United States |
| Carpenter and Moore [67] | 178 | 12.73 | Choice Theory | Survey | United States |
| Olsen [11] | 173 | 13.31 | Satisfaction and Repurchase Loyalty | Survey | Norway |
| Sichtmann [68] | 157 | 12.08 | Trust Model | Survey | Germany |
| Fullerton [69] | 151 | 10.07 | Satisfaction–Commitment–Repurchase | Survey | Canada |
| Das [70] | 142 | 23.67 | Retailer Loyalty | Survey | India |
| Pan et al. [71] | 134 | 16.80 | Customer- and Product-Related Loyalty | Review | International |
| Bao et al. [32] | 100 | 10.00 | Utilization Theory | Survey | United States |
| Toufaily et al. [72] | 96 | 13.71 | Integrative Model | Review | International |
| Anderson et al. [73] | 77 | 12.89 | Utilitarian and Hedonic Model | Panel Data | International |
| Van der Westhuizen [74] | 26 | 13.00 | Brand Experience and Loyalty Model | Survey | International |
| Amine [75] | 232 | 10.53 | True Brand Loyalty Construct | Conceptual | International |

### 3.3. Measure Design

The constructs of loyalty are multi-dimensional. This renders both qualitative and quantitative designs possible [37]. Having reviewed the definitions and dimensions, as well as the predictions for varying constructs, a qualitative analysis of the research design of the measures in the literature was conducted, as shown in Table 3.

**Table 3.** The research design for measuring loyalty (selected).

| | Research Design/Items/Measurement Model/Hypothesis Test | Data/Sample | Scale | Study |
|---|---|---|---|---|
| 1. Quantitative | Cronbach's alpha/CFA/PLS/SEM/path model/ECSI/2SI/direct-effects model | Face-to-face interview/online-/intercept and questionnaire survey/random sampling by call/mail survey | Likert (5/6/7/10 points)/semantic differential | Sirohi et al. [76]; Wallace et al. [22]; Palmatier et al. [62]; Sichtmann [68]; Vlachos et al. [53]; Das [70]; Park and Kim [77]; Strenitzerová and Gáňa [78]; Diallo et al. [17] |
| | Multinomial logit model/weighted least squares/linear regression/meta-analysis/multivariate regression | Panel data | Weighted/average market share/ordinal | Cunningham [10]; Day [14]; Tellis [55]; Dekimpe et al. [79]; Pan et al. [71] |
| | Descriptive statistic/content analysis | Literature review/personal interview | Frequency/distribution/5-point/percent/rating | Brown [80]; Wiese et al. [81]; Toufaily et al. [72] |
| | Naturalistic inquiry | Semi-structured depth interview | | Gremler and Brown [12] |
| | Transcripts and content analysis | Focus group interview | Continuous scale | Huddleston et al. [82] |

**Table 3.** *Cont.*

| Research Design/Items/Measurement Model/Hypothesis Test | Data/Sample | Scale | Study |
|---|---|---|---|
| Behavioral loyalty | | | Anderson and Srinivasan [42]; Srinivasan et al. [83]; Ailawadi et al. [64] |
| - I can easily choose another brand, if my preferred brand is not available in the supermarket; | | | |
| - I prefer the brand I always buy instead of trying another one that I am not sure about; | | | |
| - Once I choose a brand, I do not like to switch. | | | |
| Attitudinal loyalty | | | Yoo and Donthu [84]; Das [70] |
| - I consider myself loyal to the store; | | | |
| - I will not buy products from other retailers if I can buy the same item at the store; | | | |
| - The store would be my first choice. | | | |
| Cognitive, affective, conative, and action loyalty/composite loyalty | | | Harris and Goode [85]; Oliver [9]; Palmatier et al. [62] |
| - I would continue to buy this brand from this company even if prices were increased somewhat; | | | |
| - This company's prices are reasonable considering the value I receive; | | | |
| - I feel that I am getting a good deal in my dealings with this company. | | | |

Note: CFA, confirmatory factor analysis; PLS, partial least squares; SEM, structural equation modeling; ECSI, European Consumer Satisfaction Index; 2SI: two-step single-indicant estimation method.

*(Left margin label: 2. Items/Questions)*

Table 3 shows the types of research designs employed in the existing studies; examples of questionnaire items, methods, or data sources; measurement scales; and studies. Given the diverse designs of the studies exploring the antecedents and predicting the outcomes of loyalty theories, constructs are empirically divergent [71]. However, the relationships among variables are under one of the following three dimensions: behavioral, attitudinal, and cognitive [66]. As dimensional constructs, researchers often use factors that consist of indicators. A variety of model analyses have been conducted so far, including commonly used methods such as confirmatory factor analysis (CFA), partial least squares (PLS), and structural equation modeling (SEM). Some other methods have not been frequently used, such as naturalistic inquiry and content analysis in qualitative studies, multinomial logit models by panel data analysis, the European Consumer Satisfaction Index (ECSI), the two-step single-indicant (2SI) estimation method, and the direct effects model. The CFA tests the consistency of the measures of constructs in empirical studies that specify one or more latent variables with a fixed scale [86] and is traditionally employed instead of confirmatory composite analysis, which is a better fit when including emergent variables. The next commonly used model analysis is PLS-SEM, which enables researchers to estimate complex models with many constructs. The most used measurement scale is the Likert scale. The results of this review show that the research designs of the papers analyzed differ according to the data, sample, and dimensions of the variables. Only two studies used a qualitative design. One of them conducted a transcript and content analysis [82] and tested the relationship between the antecedents of relative attitude and repeat patronage using the model of Dick and Basu [16]. The other study involved a theory development for service loyalty [12]. In the beginning phase of theory construction and question formulation, in-depth interviews were applied to determine how service loyalty can be constructed and what antecedents are important for both customers and managers. The results showed that service loyalty is a multi-dimensional construct. Therefore, based on this revelation, subsequent empirical studies can be conducted.

Antecedents of loyalty were hypothesized in some empirical studies as the main constructs of the relational concept framework, while other studies investigated these antecedents at the moderator or mediator levels [14,87]. We review them subsequently within the structure of loyalty taxonomy by combining the three dimensions with the two relational levels (main construct and moderator or mediator) and the sustainability level, which has not been hitherto fully explored.

### 3.4. Antecedents and Dimensions at the Determined Construct Level

Despite typological differences, the main constructs that lead to loyalty may follow the same dimensions. Even so, the same antecedents may have more direct and determined effects on outcome loyalty in some studies than in others.

### 3.4.1. Brand Loyalty

Measures capturing both the attitudinal and behavioral dimensions are recommended for measuring brand loyalty. A two-dimensional concept was developed by Day [14], who queried the loyalty measures of purchase response. The single dimensional variable alone could not identify the difference between intentional loyalty and the "spurious" loyalty related to the long-term purchasing of a brand [88]. Therefore, Day [14] proposed four constructing factors: sociodemographic, price and store response, exposure to information, and reaction to the purchasing environment [67]. Indicators underlying the attitudinal dimension, such as economic consciousness and confidence of judgment, were also formulated. The outcome focused on consumers' brand preferences, which represented brand loyalty [89–93]. Since then, the exploration of the relationship between both dimensional constructs and brand loyalty, as well as the interrelationship between them, has ensued [16,80]. Satisfaction is regarded as one of the key behavioral constructs for developing brand loyalty [9,16,94–97]. It is based on product usage. A weakness of this argument is its insufficiency in detecting other mechanisms that influence consumers' fortitude, such as switching costs at the beginning of its formation. Additionally, subjective norms and social bonding, which represent the degree to which social group support affects consumers' decision to retain their loyalty, were ignored. The perceived value and perceived product quality, as attitudinal dimensional constructs, were proven to be significant in predicting brand preferences that led to brand loyalty [38,98,99]. As such, they were discussed merely in terms of being related to product brands. Brand trust is often hypothesized to be related to the degree of commitment predisposition toward a brand [45,49,59,65,69]. It is related to both behavioral and attitudinal loyalty and is highly valued, as it creates exchange relationships [35,36,68].

### 3.4.2. Service Loyalty

This type of loyalty is an extension of brand loyalty and was developed for service organizations that provide somewhat intangible products [12]. The core construct of service loyalty is service quality [48,100], which is a multi-dimensional concept that includes reliability, responsiveness, assurance, empathy, and tangibles. Thus, it emerged that attitudinal quality dimension performance has a varying influence on service loyalty by industry [9].

### 3.4.3. Store Loyalty

Affective and conative antecedents are more predictive of food store loyalty than the cognitive dimensions. Satisfaction as a cognitive dimension does not imply food store loyalty beyond social norms [82]. A contrary finding was obtained by [101], who argued that store loyalty may increase with customer satisfaction. Additionally, some studies argued that store loyalty is a source of retailer loyalty, although retailers are categorized as service providers [102]. Therefore, service/merchandise quality is regarded as the main determinant of store loyalty. Simultaneously, the perceived value of the focal store and value for money were empirically proven to be significant determinants of store loyalty [7,103,104].

### 3.4.4. Retailer Loyalty

Retail loyalty can be understood as loyalty at the organizational level (corporate level) or the retail chain level. Its antecedents can be the characteristics of a brand, service, and store loyalty, but retail loyalty can also be considered at the corporate level [105]. Demonstrating corporate social responsibility, for example, by offering organic products at

food retail chains, is regarded as a practice that engenders food retailer loyalty [58,106]. The effect of customer satisfaction on the retailer loyalty hypothesis was confirmed as positively significant. Brand value was empirically demonstrated as an essential component in the formation of retailer loyalty [107]. In accordance with the retailer format, such as retailers' websites or presence on a social media platform, information access, and experiential shopping were verified as having a causal relationship with retailer loyalty [73].

### 3.5. Moderator and/or Mediator Level

A mediator may intervene and reveal the true relationship between two related constructs (antecedents and loyalty), while a moderator may change the strength or direction of a relationship between two constructs in a hypothesized model [108].

### 3.5.1. Brand Loyalty

Although satisfaction was adopted in many empirical studies as the main construct leading to loyalty, there is a distinction between "manifest satisfaction" and "latent satisfaction", according to Bloemer and Kasper [44]. A strong significant effect of "manifest satisfaction" on true loyalty has a moderator effect on the relationship between satisfaction and loyalty. In this case, satisfaction can be measured under the attitudinal and affective dimensions. Another affective dimension that has been researched as a mediator is affective commitment, which has been shown to mediate the link between brand experience and brand loyalty [61,63]. Furthermore, the attitudinal dimension of brand value also plays a primary role in influencing brand loyalty at the moderator level. Brand value can be perceived by customers. This moderating effect has been elaborated by several scholars [87] in terms of affecting brand loyalty. Additionally, perceived quality is considered as a moderator of the relationship between satisfaction and loyalty [109]. In the consumer research field, simulated personal characteristics, such as age, income, and variety seeking, were also explored as moderators [88]. These variables significantly moderate the relationship between satisfaction and loyalty [46].

### 3.5.2. Service Loyalty

Satisfaction is another mediator linking service quality and service loyalty [39]. Some studies claimed that brand experience mediates the association between self–brand connection and service brand loyalty [74]. Nevertheless, others asserted that brand trust mediates the relationship between brand experience and service brand loyalty [110]. The latest research studies show that brand trust positively mediates the relationships between brand image and loyalty types [17,62].

### 3.5.3. Store Loyalty

Consumer satisfaction is disputable as a moderator for defining store loyalty. It was shown to decrease if the number of corresponding manufacturer brands was reduced in retail stores [101]. Other studies argued that store brands build store loyalty directly [52,111–113]. One point of consensus is that behavioral measures alone are inadequate, either as major constructs or as mediator variables [21].

### 3.5.4. Retailer Loyalty

At the corporate level, trust is an important mediator of retailer loyalty [58,84] and a source of it [114]. Due to the variety of retailer attributes, as well as channels and formats, the factors of multi-channel employment, service outputs, portfolios, and satisfaction affect retailer loyalty [22,47,83]. Retailer service outputs are mediated by satisfaction and positively affect retailer loyalty.

### 3.6. Untapped Loyalty at the Sustainability Level

At the sustainability level, the discussion is not limited to the present; rather, it is future-oriented. While consumers can be intrinsically loyal to a brand, store, or retailer at the

chain level, they are also potential switchers [79]. In marketing, sustainability has not been given a unified definition used to build models and conceptualize relationships [20,78]. Thus, dichotomous results are evident in the literature on consumer behavior. One category of results is merely environment-focused, which indicates the perception of green or sustainable products, while the other is holistically defined, in relation to which how it affects consumers' preferences [115] and their loyalty under a multi-dimensional framework with emergent factors has not been explored. Long-term sustainability loyalty requires the support of lifetime customer value (LTV) according to Schultz and Block [33]. They proposed a brand sustainability concept that has not been constructed, measured, or evaluated. This concept is beyond brand loyalty and involves the next level of brand growth. In this study, the suggested measures are based on the organizational-level average growth rate (AGR), which is the main indicator of brand sustainability. In fact, organizational profit is a sustainability value. Thus, understanding what values consumers appreciate and where managers should direct their attention to achieve a marketing edge is vital [99,116].

The relationships among the four core values of sustainability and consumer brand loyalty may not be measured solely by corporate profit. Retailers who exhibit core sustainability values can acquire loyalty over the long term [117]. The discussion on "retailer as the brand" has persisted for almost two decades [118,119]. Notwithstanding, the loyalty measures at the sustainability level lag and are fragmentary. On the one hand, retailers distinguish among product, intangible service, and restricted store brands. On the other hand, brands possess all the above-mentioned characteristics. A satisfying relationship between retailers and consumers, rather than a passing transaction, should be built [120]. Consumers expect retailers to commit to environmental and social value creation [53]. Some researchers categorize loyalty into food brand loyalty, namely, the sustainable products offered by retailers, such as organic food. The analyzed studies suggested that this type of loyalty may be facilitated by attitudinal measures since the purchase of ethical and sustainable products reflected a strong attitude toward those brands that represented consumers' individual values [121–124]. This motivated consumers to buy more organic food than others [125–127]. Studies have shown that a single-dimensional measure fails to conceptualize retailer loyalty. Thus, researchers and managers should consider multi-dimensional measures of loyalty for food retailers that integrate the core values of sustainability. However, the starting point varies. The most argued sustainability-related keyword in retail literature is organic, followed by sustainable/sustainability, green, environment/environmental, carbon footprint/$CO_2$, and CSR/social responsibility, in this order [81,128–130]. Organic branding is a marketing strategy of food retailers that contributes to organic growth based on economic, social, and environmental values. Both the attitudinal and behavioral dimensions are salient for food marketing and have been demonstrated to reliability predict and positively influence consumers' true loyalty in the long term [131]. The emergent factors include the public policy, labeling scheme, and global sustainability movement which are arising and enforced to influence consumers' loyalty to retailers.

## 4. Results

Following the integrative review, the results of this study are as follows. Firstly, food retailer loyalty toward sustainability can be defined not only from product-related dimensions but also from service-dominant scopes. The attitude to product quality cannot be the single element that generates consumers' loyalty; service-dominant retailers can influence patrons and/or acquire new consumers by retailing service quality. To define loyalty, one should consider the special characteristics of the food retail sector. This result concurs that food retailers possess two attributes, in that they supply goods and provide services. Secondly, consumer satisfaction and trust are arguable either as the main construct or as moderator and mediator among the constructs to form the consumers' loyalty. These two constructs can be derived from attitudinal, cognitive, and behavioral dimensions which require appropriate combined methods and measure items for their examination relative to consumers. Thirdly, there is a lack of an integrated framework for emergent

sustainability values based on sustainable products and services. The newness of this study enriches the existing relational concept between attitudinal and behavioral loyalty by integrating emergent sustainability values in the business practice.

## 5. Discussion

Promoting consumer loyalty to retailers relies on dynamic long-term marketing inputs. To effectively achieve consumer-oriented brand loyalty, the identification of the antecedents and dimensions of sustainable food marketing is necessary. Our systematic review suggests that there are indicative distinctions across the literature, substantiating diverse forms of customer loyalty, such as brand, service, store, and retailer loyalty. However, there are also inconsistent antecedents and sequences of measures in predicting the overall loyalty framework. The traditional agreement between the attitude and behavior dimensional levels is unresponsive to sustainability and its inherent values. A variety of effects of both the determined constructs and mediators or moderators were also inconsistent regarding the final loyalty construct.

### 5.1. Theoretical Contributions

The theoretical contributions of this study can be divided as follows:

(1) This review discerned that the dimensional research gap in relation to consumers' cognitive concerns is represented by the lack of product/service life cycle in consumption practice. For example, in predicting loyalty, the behavioral measure fails to forecast the pre-purchase decision-making process, as a pure attitudinal measure may not capture actual purchases [132,133]. Furthermore, mixed measures at different construct levels may not reflect the direction of the actual causal relationships. Consequently, there is a lack of integrated dimensions to effectively predict brand loyalty.

(2) The measures of brand loyalty are suggested to be improving according to the industrial and marketing focus. We observed that the mediator effects vary across industries and service settings. Thus, it is impossible to use the same definitions and measures for loyalty in diverse industrial categories. The dimensions of satisfaction, brand value, and trust were measured in some studies as exogenous constructs, while these dimensions served as mediators in others. Product brand measures should not be used to predict service brands. Food retailers sell goods, while also providing services. Their sustainable branding activities evoke consumers' cognition and increase their value. Therefore, consistent, transcending, and dynamic factors should be developed based on consumers' perceptions.

(3) Regarding the data, sample, and research phase, qualitative and quantitative methods lend themselves to three research directions. Qualitative research aims to stimulate new theory development beyond the existing dimensions of the loyalty framework [134]. Quantitative research tests the reliability, validity, and significance of relationships among the antecedents of loyalty. Finally, field and consumption practice research can more directly observe consumers' actual behaviors in association with their loyalty across the product/service cycle.

(4) Further research on retailer brand loyalty can be extended to study the relationship among service-dominant loyalty, environmental consequences, and social well-being. In this case, a mixed qualitative-quantitative approach may be appropriate to find the emergent factors in relation to sustainability and identify the complex correlations, thus producing enlightening results.

(5) The conceptualization of consumer loyalty for retailers may integrate constructs involving emergent factors and elements of sustainability value, sustainable marketing elements, and loyalty. Organic marketing, innovativeness of store formats, and improvement of healthy and nutritional food products and services stimulate organic and retailer brand growth [135,136]. The value created by this growth can contribute to loyalty construction in coordination with economic, environmental, consumer, and social values. Previous research studies concur that consumer attitude toward

sustainability is related to consumer loyalty if brands make strategic decisions that have positive impacts on the environment [137,138]. This research call is consistent with research studies arguing that perceived value is an important antecedent of brand loyalty. The emergent food policy may strengthen the perceived trust of food retailers in association with food safety and sustainability values.

(6) This research study is limited in the scope of constructs and measurements of loyalty in the food retail sector. The other limitation is that it considers only antecedents and effects on loyalty in the B2C background but not for B2B.

### 5.2. Managerial Implications

The main managerial contribution of this study is bridging the dimensional gap between theory and practice by applying a sustainable marketing framework to enhance retail brand loyalty. As loyalty is often regarded as a multi-dimensional construct, consumers can switch quickly to competitors and content can be changed dramatically because of the dynamic social and environmental influences. This study suggests that retail corporate managers should identify both loyal consumers in terms of their purchase behavior and unexplored consumer groups that attitudinally and cognitively lean toward retailers' sustainable marketing development. They can also strengthen loyalty by emphasizing positive organic business growth and organic knowledge diffusion and by further guiding the specialized organic marketing efforts toward sustainability. Using a simpler but strategic value measure may sustain loyalty in the long term.

## 6. Conclusions

The purpose of this study is to discuss the current knowledge about consumer loyalty in the food retail sector among influential marketing research studies and to understand the critical dimensions and constructs around the concept of sustainability. It grasps the power of sustainable business concepts, such as organic growth by branding strategy of retailers, product/service lifecycle values, consumer satisfaction, experiential brands, and consumer trust, in utilizing consumer loyalty to the benefit of retailers. Regarding the role of food retailers as suppliers of goods/products and service providers, conceptualizing and constructing consumer loyalty should be considered from both perspectives bearing in mind sustainability values.

This study focuses on the antecedents, research design, and measures of loyalty and offers suggestions for further relational research on sustainability. However, the variables and items used to construct the effective relationships between sustainability-based food branding and consumer loyalty must be thoroughly discussed in future studies. The potential areas of improvement in further research studies can be developed in three directions: (1) An integrative framework including the design of emergent variables derived from sustainability values should be developed. (2) The relationship between sustainable marketing and satisfaction can be explored by the consumer practice method in order to detect manifested, latent, and emergent satisfaction. (3) Studies that focus on the measurement of consumer values, corporate social responsibility, and sustainable well-being are needed to examine the effects of consumer loyalty on food retailers.

**Author Contributions:** Conceptualization, Y.T. and Q.K.; methodology, Y.T. and Q.K.; formal analysis, Y.T.; resources, Y.T.; data curation, Y.T. and Q.K.; writing—original draft preparation, Y.T.; writing—review and editing, Y.T.; visualization, Y.T.; supervision, Q.K.; project administration, Y.T. All authors have read and agreed to the published version of the manuscript.

**Funding:** The authors received no external funding.

**Institutional Review Board Statement:** Not applicable.

**Informed Consent Statement:** Not applicable.

**Conflicts of Interest:** The authors declare no conflict of interest.

# Appendix A

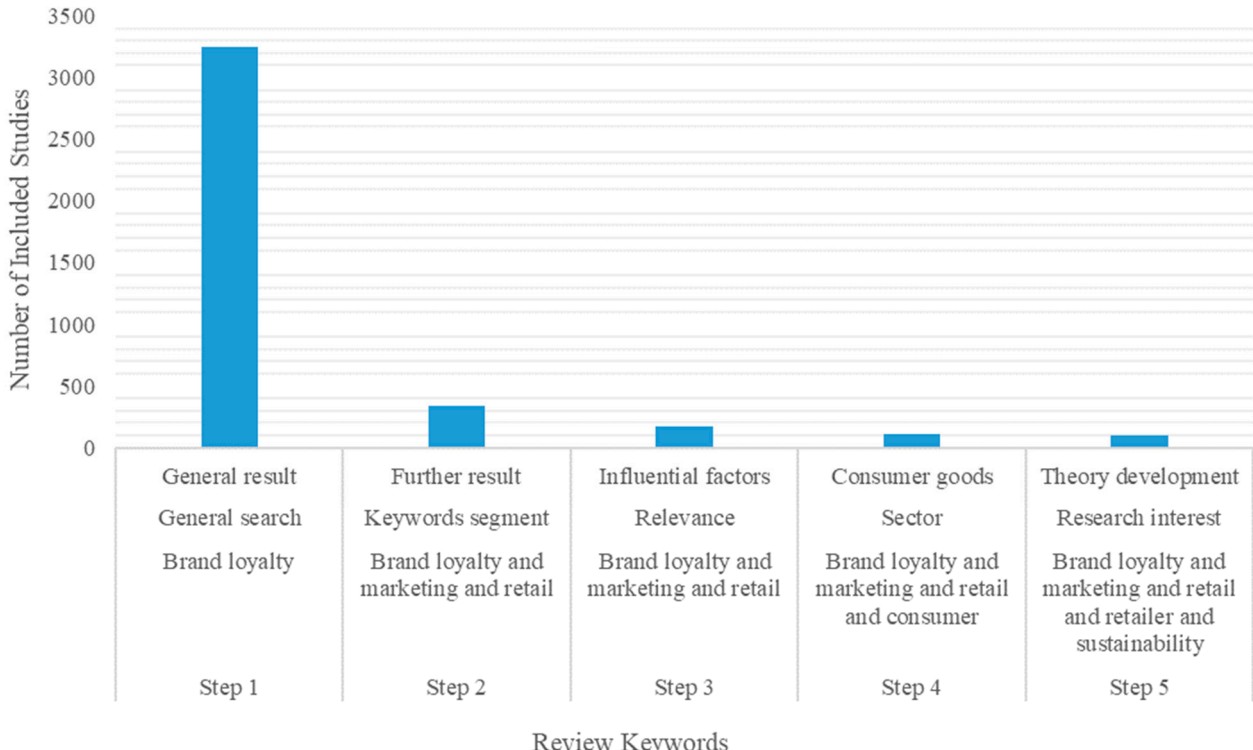

**Figure A1.** Review steps with keywords.

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
