# Peer review of "A Review of Antecedents and Effects of Loyalty on Food Retailers toward Sustainability"

_sustainability, doi:10.3390/su132313419_

Round 1

Reviewer 1 Report

Dear Authors I sincerely congratulate you for your work. I am positively impressed by the natural flow of arguments regarding the concept of loyalty, its various hypostases and the connections identified with sustainability in the field of food trade.

 - The paper is very well organized and the information is clearly structured.

 - The analysis stages regarding the systematic review are clear and well intercorrelated.

 - Pay attention to the format of figure no2, as the right margin is out of the page. It can be easily corrected moving all the boxes slightly to the left.

 - As recommendations I advise authors to be more specific and clear on some future directions of research, maybe to have an opinion regarding possible connections already identified and the method (type) of research that can address accordingly these connections (for example a mixed qualitative-quantitative approach may lead to some interesting findings.... (I particularly appreciated the conclusions from point 4.1. (3)).

Good luck!

Reviewer 2 Report

Dear authors,

a very good paper, just please be careful to the formatting.

Author Response

Thank you very much for the encouragement.

For the formatting, I have submitted to the Editing Service Team for reformatting.

Reviewer 3 Report

The authors have taken a very vibrant topic, which has significant theoretical and practical implications. The authors have put their best efforts to execute this review paper. However, I have following reservations and suggestions to the authors:

1) The authors should add the practical and theoretical implications in the end of their abstract as well.

2) The motivation, significance, objectives and novelty of the review paper should be discussed at the end of "Introduction" section.

3) In methodology section the authors did not mention the secrutinizing of theories, thus, it is recommended to the authors to mention step number 2, to analyse the theories, which were used in papers that were included in this review paper, On page 3 just below the Figure 1. 

4) The Review of literature is very comprehensive and has taken sufficient research papers. The only recommendation is to write the review in an audit form and try to link the objectives of the review paper.

5) The discussions section should be very critical, because this section provide you the opportunity to sell your conception and ideas to your readers. This is quite weak, I will recommend to separate conclusion from the discussions. The discussions should be presented the novelty of your review paper, what you have deduced and what new knowledge you have provided to the academia and industry.

6) It is better to add one more section before discussion in which you address the outcomes, and newness or new conceptual model, which you have extracted from this review paper.

7) The authors have mentioned that they have taken paper from 1961 to 2018, however, authors also considered the papers of 2020, and 2021. Thus, it is better to indicate the period from 1961 to 2021. In this way the scope of the review paper will be more broader. 

8) Conclusion should be separated from the discussions, and conclusion should be one step ahead of your findings. 

9) Figure 2 should be adjusted because it is beyond the page boundaries. it could be displays in a landscape form.

10) In the end the limitations, and of the review paper should also be discussed, and also suggest the potential areas of future research studies.

Reviewer 4 Report

Paper strengths: although this is not a new/hot topic, it is relevant and contributing to study loyalty from the perspective of sustainable consumption. The paper is well written.

Suggestions for improving the article are specified below, section by section.

Abstract: the abstract is well constructed, with objective, method, results and academic contributions of the study. However, I suggest the authors to present, at the beginning of the summary, in a synthetic way, the gap(s) in the literature that justify the relevance and opportunity of the work (i.e., studying loyalty in the context of food retailing towards sustainability is relevant, but is there a real gap in the literature? Have other studies in the literature been carried out with a similar objective? If so, where does this one advance?). In addition, I definitely recommend closing the abstract not only with academical contributions, but moving on to practical implications: how the results of this review can inspire managers working on food retail to improve customer loyalty?

Introduction: I think that the introduction should present the context of the research theme, especially regarding gaps in the literature, resulting in the proposition of the research question and objectives. The gap that was not highlighted in the abstract has now been presented, good job! Questions to be clarified: 1) The authors need to decide whether the focus of the article is on organic marketing or on food retailers. Although they are conceptually distinct terms, they are used interchangeably in the research questions and in the study objective (page 2), which is not adequate. 2) The following statement is from which authors? “However, to a certain degree, consumers’ purchase intentions and their perception of retailer brands coincide with their knowledge of whether the brand prioritizes values of sustainability” (page 2). 3) I still miss managerial implications. 4) Importantly, the article does not have a theoretical background section, so that we can check whether the state of the art in the researched topic has been properly portrayed.

Concerning this issue, the theoretical framework should be composed, in its majority, by recent scientific papers (from the last 5 years or, if possible, from the last 3 years), which is important to portray the state of the art in the theme. In this sense, both recentness and scientificity are fundamental. Thus, books, reports, internet references and working papers should be avoided as much as possible. Although the authors have privileged references of scientific articles (95%), 119 of the 125 references are from scientific articles, an excellent number!), only about 13% of these references (16 of 125) date from the last 5 years (2017-2021). Although scientificity is very good, recentness needs to be improved, because such a current issue as loyalty on sustainable environments cannot be stagnant in time. Thus, I’m afraid the theoretical support to develop the research questions does not seem adequate and sufficient. This is an important weakness of the article!

Methodology: First of all, I suggest not calling this section Methodology, because methodology is the science that studies the various methods available and this section of the article specifically addresses the methods adopted by the authors to achieve the study objectives. So, Methods is definitely a better term to name this section. There is an epistemological adjustment of the methods chosen in the light of the proposed objective, it’s very good! However, there are questions I would like to ask for reflection: 1) why choose Web of Science and the 1961-2018 period? 2) If we are at the end of 2021, why did the search end in 2018? Each methodological choice needs to be justified. 3) Isn't marketing a too broad term to be used in the initial research phase? 4) Relevance is usually a step in the review protocol where only articles published in high-impact journals (e.g., JCR>1.0) are selected and not the fact that the context is B2B, as reported in Appendix A; this criterion would be more related to eligibility. Please explain better. Also, nothing was mentioned in this session about qualitative methods of analysis; the focus was only on the systematic review steps. Furthermore, I suggest presenting the figures and tables after introducing them into the text, not on this section only, but on the entire article.

Literature Review: This section should be called Results and not Literature Review. Although the results are well organized, I wonder why the authors didn't group them in order to answer the 3 research questions proposed on the Introduction. The relevance assessed by the number of citations per year was used in the presentation of results, but ideally it would have been used as a filter in the selection of articles to compose the review corpus. Please, pay attention to this sentence on page 9: “….PLS-SEM which enables including higher-order constructs in the model”, because CFA also does so. Besides, as I pointed out in the Methodology section, I’m afraid I was unable to identify the use of qualitative methods to analyze the results of the review carried out because the authors only described them. No efforts were made to define new categories of analysis, nor signaling future directions of investigation. The topic 3.6 (Untapped Loyalty at the Sustainability Level) was the best one in this section.

Discussion/Conclusions: this section should be much more developed, promoting a dialogue between the engendered results and the theory visited (but as the article does not have a theoretical background section, this dialogue has been compromised). What the authors called Theoretical Contributions, in fact, is an agenda for further studies, which is very important because it is one of the main results expected from a systematic literature review. My suggestion in this regard is that the authors better explore this agenda, going beyond the agendas proposed in the studies that comprised the review and developing their own agenda. The theoretical advance that the study brought is not properly highlighted either. Are there other reviews on loyalty from the perspective of sustainable consumption? If so, they need to be mentioned and the difference of the article in relation to these other studies, which really constitutes theoretical contributions, presented. I missed the limitations of the study, which were not pointed out. Finally, managerial implications emerged, but they are still timid and should be less general and more specific for food retail managers.

I wish the authors success in future submissions!

Round 2

Reviewer 4 Report

Unfortunately, many of the points I raised as important weaknesses of the article in the first round of reviews have not been resolved. I highlight them again below.

Introduction: The authors justified the absence of a Theoretical Background section because it was a review article. It’s not satisfactory. Literature review articles also have a Theoretical Background section. Concerning the recentness of the references, the authors added 3 new references from the last 5 years only raising the percentage from 13% to 15%, which is still very low. That is, the theoretical support to develop the research questions remains inadequate and insufficient. Once again, the state of the art in the researched topic hasn’t been properly portrayed. Thus, this important weakness of the article still remains.

Methods: the authors have not yet explained why the choice of the 1961-2018  period of analysis. If the survey started in 2019, as pointed out by the authors, we are already in 2021 and the articles research period needs to be updated. This update is essential in literature review articles. What factors did the authors use for the review protocol's relevance criteria? Once again, nothing was mentioned about qualitative methods of analysis; the focus was only on the systematic review steps throughout the article.

Literature Review: In my opinion, it didn't make much sense to create a Results section of just 1 paragraph and continue with the Literature Review section. Furthermore, I still do not identify the use of qualitative methods to analyze the results of the review carried out because the authors only described them. I registered this same concern in the previous paragraph as well.

Discussion/Conclusions: the limitations of the study are still missing. Also, the agenda for future studies should not be at the Conclusion, but should form a separate subsection in the Discussion, as I pointed out before.
